# Cingulate cortex shapes early postnatal development of social vocalizations

Gurueswar Nagarajan[1], Denis Matrov[1], Anna C Pearson[1], Cecil C Yen[2], Sean P Bradley[3], Yogita Chudasama[1,3]*

[1]Section on Behavioral Neuroscience, National Institutes of Health, Bethesda, United States; [2]NeuroImaging Facility, National Institutes of Health, Bethesda, United States; [3]Rodent Behavioral Core, National Institutes of Health, Bethesda, United States

## eLife Assessment

This **important** study investigates the influence of the cingulate cortex on the development of the social vocalizations of marmoset monkeys by making bilateral lesions of this brain area in neonatal animals. The evidence supporting the authors' claims is **convincing**. The work will be of broad interest to cognitive neuroscientists, speech and language researchers, and primate neuroscientists.

**\*For correspondence:**
yogita.chudasama@nih.gov

**Competing interest:** The authors declare that no competing interests exist.

**Abstract** The social dynamics of vocal behavior have major implications for social development in humans. We asked whether early life damage to the anterior cingulate cortex (ACC), which is closely associated with socioemotional regulation more broadly, impacts the normal development of vocal expression. The common marmoset provides a unique opportunity to study the developmental trajectory of vocal behavior and to track the consequences of early brain damage on aspects of social vocalizations. We created ACC lesions in neonatal marmosets and compared their pattern of vocalization to that of age-matched controls throughout the first 6 weeks of life. We found that while early life ACC lesions had little influence on the production of vocal calls, developmental changes to the quality of social contact calls and their associated sequential and acoustic characteristics were compromised. These animals made fewer social contact calls, and when they did, they were short, loud, and monotonic. We further determined that damage to ACC in infancy results in a permanent alteration in downstream brain areas known to be involved in social vocalizations, such as the amygdala and periaqueductal gray. Namely, in the adult, these structures exhibited diminished GABA immunoreactivity relative to control animals, likely reflecting disruption of the normal inhibitory balance following ACC deafferentation. Together, these data indicate that the normal development of social vocal behavior depends on the ACC and its interaction with other areas in the vocal network during early life.

## Introduction

Vocal behavior is a critical mediator of social communication through different life stages of many animals, and particularly in social species such as primates (*Eliades and Miller, 2017*). The common marmoset is a small, arboreal monkey with an elaborate repertoire of acoustic calls. While the meaning and usage of most marmoset vocalizations are not well understood, research has shown that different call types convey information about their social organization, environment, and the presence of food or predators (*Oliveira and Ades, 2004*, *Eliades and Miller, 2017*). Moreover, these calls undergo developmental progression. During the first postnatal months, the acoustic properties and usage of marmoset infant vocalizations change markedly. For example, for different call types, parameters such as duration and frequency follow typical trajectories during the first months of life, transitioning

from an immature babbling phase with a mixture of proto-calls to a more discrete and contingent usage of adult-like calls (*Egnor and Hauser, 2004*; *Takahashi et al., 2015*). Recent evidence also suggests that parental or social interaction plays a significant role in the proper development of normal vocal behavior, raising the prospect that important aspects of marmoset vocal behavior are learned (*Gultekin and Hage, 2017*; *Gultekin and Hage, 2018*). Thus, a failure to convey appropriate social information through vocal calls could potentially influence how the infant develops its social interactive skills.

In this study, we focus on the anterior cingulate cortex (ACC) and its contribution to vocal behavior and its development in early life. The ACC is a limbic cortical region known to contribute to vocal behaviors (*Jürgens, 2002*, *Miller et al., 2005*), and particularly those associated with emotional states (*Jürgens and Pratt, 1979*; *Sutton et al., 1981*; *Kirzinger and Jürgens, 1982*). Electrical stimulation of the most rostral segment of the ACC elicits vocalizations (*Jürgens and Ploog, 1970*; *Müller-Preuss et al., 1980*; *Jürgens, 1976*), whereas ACC ablations limit spontaneous vocalizations (*Aitken, 1981*) and voluntary control of vocal behavior (*Sutton et al., 1974*; *MacLean and Newman, 1988*). Its dense anatomical connections with the amygdala (AMY) (*Vogt et al., 1987*; *Hoesen et al., 1993*) underscore its role in shaping the affective component of vocalizations (*Vogt and Barbas, 1988*; *Lloyd and Kling, 1988*; *Aggleton, 1993*; *Gabriel et al., 1980*). At the same time, its descending projections to the periaqueductal gray (PAG) (*Müller-Preuss and Jürgens, 1976*; *Hardy and Leichnetz, 1981*) endow the ACC direct control over activating the brainstem vocalization pathway (*Jürgens, 2002*; *Jürgens, 1994*; *An et al., 1998*). Vocal production leads to expression of immediate early genes in the ACC (*Simões et al., 2010*), with early studies reporting that infant ACC lesions abolish the characteristic cries that infants normally issue when separated from their mother (*MacLean, 1985*). These findings implicate the ACC in volitional and emotional control over vocal output.

Longitudinal monitoring of vocal behavior provides a tractable, high-dimensional readout of the development of socio-affective circuits. It also provides a means to investigate how early life disruption to brain areas such as the ACC might affect the normal progression of social interaction. If the ACC contributes to the early-life maturation of vocal behavior, then neonatal ACC lesions should hamper the normal control of emotional vocal utterances. Here, we performed excitotoxic ACC lesions in neonatal marmosets and tracked their vocal behaviors, comparing them to age-matched controls throughout the first 6 weeks of life, and examined the impact of the early life lesion on interconnected brain regions in the vocal production network. We demonstrate that animals with neonatal damage to the ACC retained their capacity to issue calls. However, these animals showed a change in their vocal repertoire and an altered acoustic structure in their communicative 'social' calls, as well as permanent anatomical changes in the AMY and PAG.

## Results

We studied the vocal behavior in 10 infant marmosets (five males and five females) from five different sets of unrelated parents. In five of the neonatal animals, we performed surgical excitotoxic lesions bilaterally to the rostral portion of the dorsal ACC (24a and 24b) (*Figure 1A and B*, see Materials and methods). Starting 7 days before the surgery and continuing until 6 postnatal weeks, infant vocalization behavior was recorded in an isolated, temperature-controlled incubator in 5 min sessions, two to three times a week (*Figure 1C and D*). In four of the animals, the estimate of ACC volume from T2-weighted MR scans performed under anesthesia approximately 8 months of age revealed a 60% decrease in ACC volume compared with four control animals (*Figure 1E*; two animals were not scanned). Following sexual maturity, at approximately 2 years of age, the animals were euthanized, and their brains were histologically visualized to verify the extent of the lesion. We also examined downstream effects of the lesion, including an evaluation of its effects on mature neurons (NeuN), inhibitory neurotransmitters (GABA and GAD67), glial cells (GFAP and Iba1), and fiber tracts (myelin; *Figure 1F*). See Materials and methods for details.

### Verification of the ACC lesion and its impact on downstream vocal structures

The intended lesions and reconstructed ACC damage based on histological evaluation are shown in one hemisphere for four animals in *Figure 2A and B*, respectively. The ACC lesion covered most of the

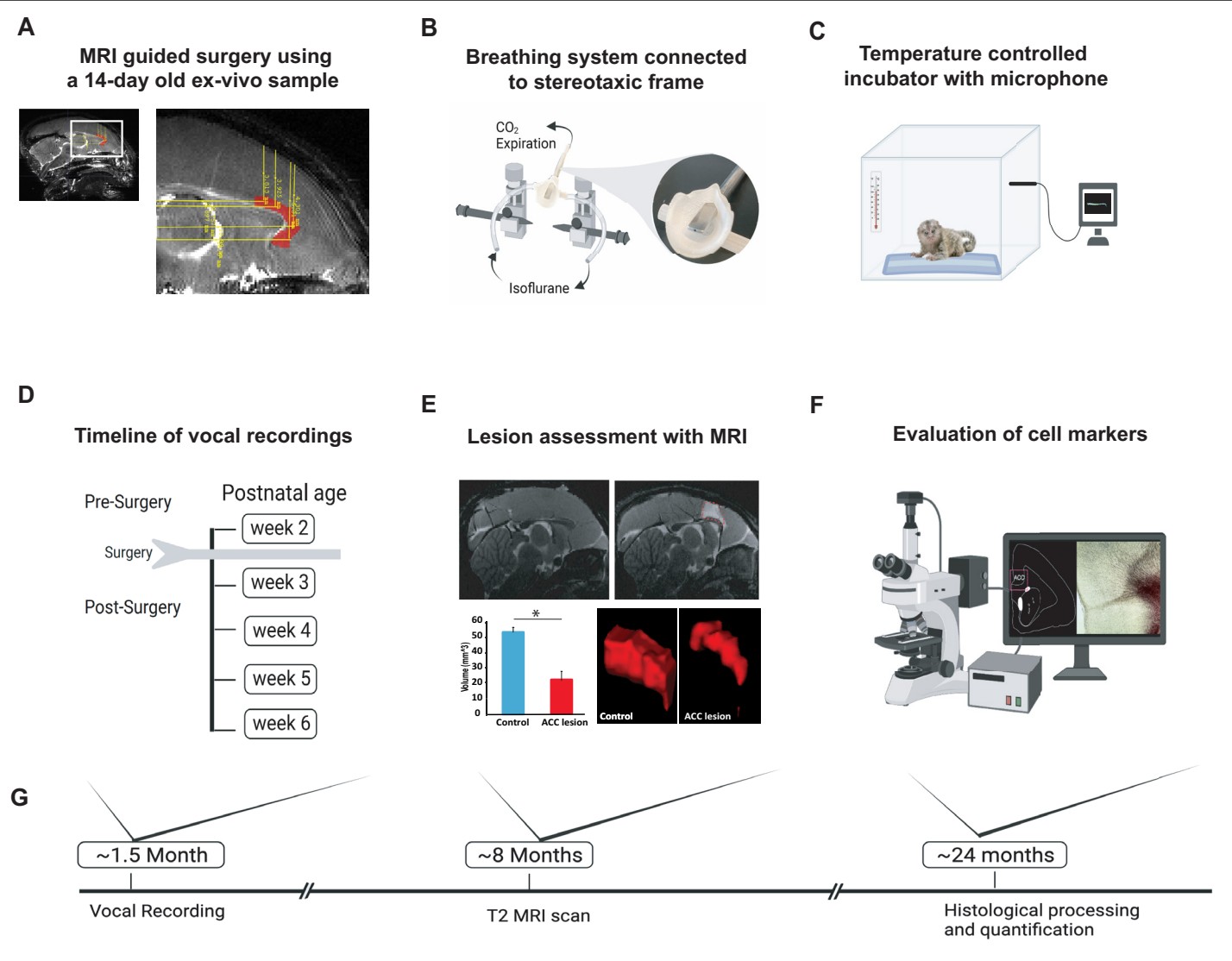

**Figure 1.** Experimental design and timeline. (**A**) Reference MR scan was obtained using an ex vivo sample of a 14-day-old marmoset. Parasagittal view of the reference scan shows location of injection coordinates targeting the rostral portion of the dorsal anterior cingulate cortex (ACC) (24a and 24b), bilaterally (red). (**B**) Gas anesthesia was supplied through a custom-made breathing system comprising a facemask fitted with a palate bar with a 0.6 mm diameter hole. The palate bar was connected to a vital monitor to accurately detect small tidal end volumes during anesthesia while the animal was secured in the stereotaxic frame. (**C**) Five-minute vocalization recordings were obtained from infants placed in a softly padded temperature-controlled incubator. (**D**) Timeline of vocal recordings from postnatal week 2 to postnatal week 6. The ACC lesion was conducted at postnatal week 2 when animals were 14–16 days of age. (**E**) Representative sagittal view of postoperative T2-weighted MR images of a control (left panel) and lesioned (right panel) infant to reveal extent of white hypersignal, which reflects edema due to injections of the excitotoxin and therefore approximate site of the ACC lesion. There was a significant reduction in total ACC volume in the ACC group relative to controls (n=4/per group; F(1,6) = 82.78, p<0.0001). A representative three-dimensional view of area 24 is presented, showing the reduced volume of the ACC (right panel) relative to the normal volume in the control (left panel). (**F**) Schematic illustration highlights the end point of the experiment involving histological processing and evaluation of cell markers. (**G**) Longitudinal timeline shows approximate age of animals following vocal recordings, MRI lesion assessments, and histological processing.

The online version of this article includes the following figure supplement(s) for figure 1:

**Figure supplement 1.** Bilateral MRI scans.

target cytoarchitectonic areas 24a and 24b of *Paxinos et al., 2011*, just above the corpus callosum. The rostral limit of the lesions was adjacent to the genu of the corpus callosum, and the caudal limit was just anterior to area 23a caudally. Dorsally, the lesions extended past 24b into motor area 6M. There was little, if any, encroachment into subgenual area 25. Apart from one case which showed some sparing of the lesion in the left hemisphere, there was extensive overlap in the placement of

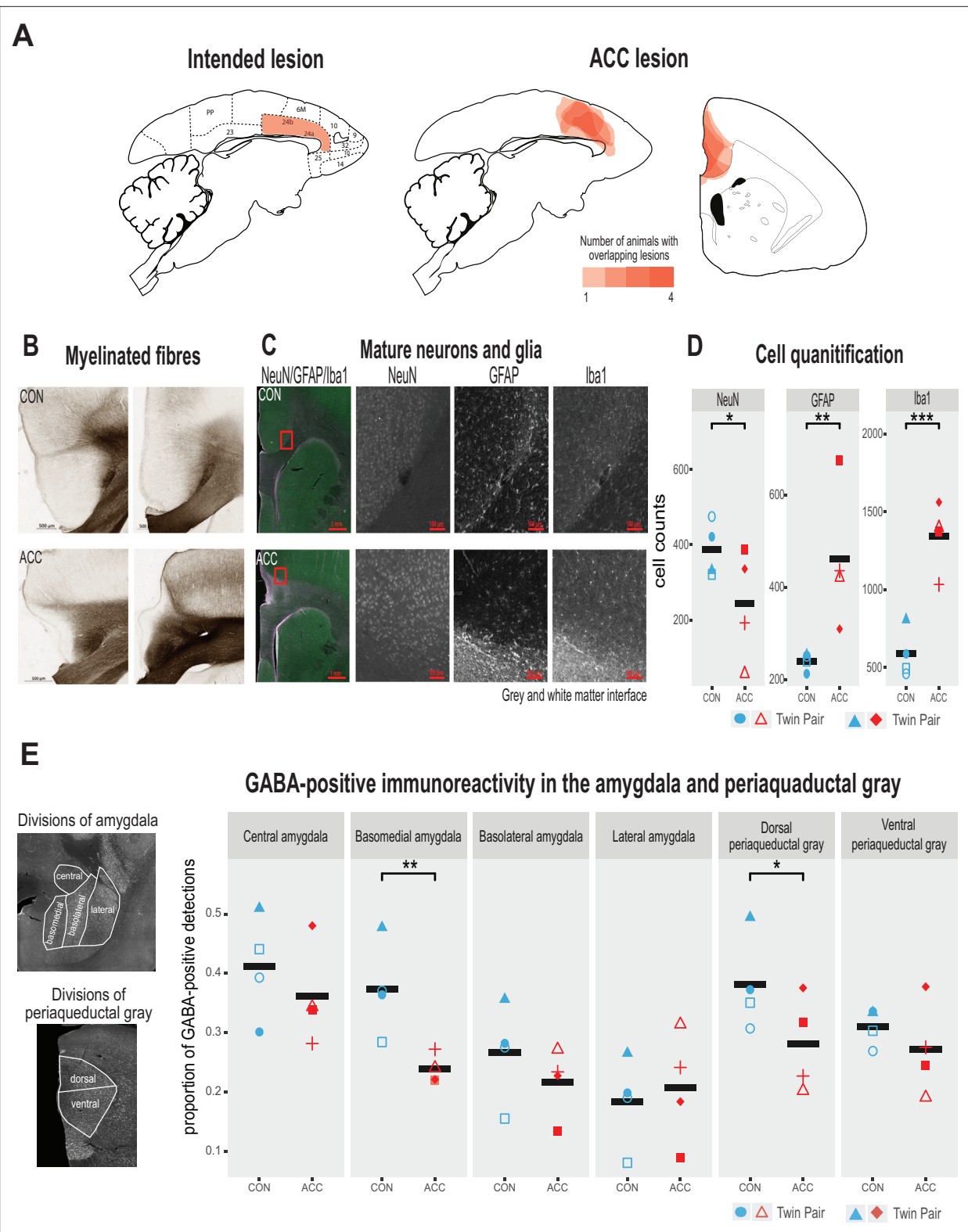

**Figure 2.** Lesion verification and impact of early life anterior cingulate cortex (ACC) lesion on vocal downstream structures. (**A**) Left panel shows a sagittal section from the standard marmoset brain depicting the intended ACC lesion shaded in red. The right panel shows schematic lesion reconstructions superimposed on a sagittal and coronal marmoset brain section depicting the extent of the ACC lesion shaded in red. Regions that appear darker indicate greater overlap in the damage present among different animals. (**B**) Magnified images of area 24 stained to visualize myelinated fibers in a representative control (top panel) and ACC-lesioned (bottom panel) animal. The normal radial arrangement of the myelinated fibers is

*Figure 2 continued on next page*

*Figure 2 continued*

disrupted following the ACC lesion. (**C**) Histological visualization of mature neurons and glia in representative control (top row) and ACC-lesioned animal (bottom row). Leftmost image shows neurons (green), astrocytes (violet), and microglia/macrophages (white) in the same image. Red square represents the magnified grayscale sections showing cell loss (NeuN), high levels of astrocytes (GFAP), and microglia/macrophages (Iba1) surrounding the lesion site at the gray and white matter interface in the ACC-lesioned animal (bottom row) relative to the controls (top row). (**D**). Histological quantification of mature neurons and glia in cortical area 24 in the ACC-lesioned animals, or in the corresponding intact cortical tissue bordering white matter in the controls. There was a reduction in the number of mature neurons (NeuN) and an increase in glia (GFAP and Iba1) in the ACC group relative to controls. $^*p<0.05$; $^{**}p<0.001$; $^{***}p<0.00001$. (**E**) Left panel shows grayscale images with anti-NeuN staining depicting divisions of amygdala (AMY) and periaqueductal gray (PAG) where relative distribution of GABA-positive immunoreactive expression was quantified. Right graphs show the proportion of GABA expression in each division depicted in the AMY and PAG. Each symbol represents one animal (ACC-lesioned animal is red, control animal is blue). Mean expression is represented by a black bar. Data for two animals in the ACC group overlap for basomedial AMY quantification. GABA-positive immunoreactivity was significantly down in the basomedial AMY and dorsal PAG.

The online version of this article includes the following figure supplement(s) for figure 2:

**Figure supplement 1.** Representative images of white matter tract from a CON and anterior cingulate cortex (ACC)-lesioned subject.

the ACC lesion. The anterior-posterior extent of the bilateral lesion for each monkey is compiled in *Figure 1—figure supplement 1* as sections from MRI images.

Representative photomicrographs of the ACC lesion and a control are presented in *Figure 2B*, which shows the distribution of myelinated fibers in the ACC region stained using a high-resolution Black-Gold II myelin stain (Histo-Chem Inc, Jefferson, AR, USA). There was clear evidence that the lesion created a major disruption to the normal radial arrangement of the fibers in the ACC region caused by extensive demyelination of the axons (*Figure 2B*). The ACC lesion also impacted the integrity of white matter tracts local to the site of the lesion (*Figure 2—figure supplement 1*), but the transverse diameter of major fiber tracts, namely the corpus callosum and the anterior commissure, did not differ between the groups. The loss of neurons, however, and the respective increase in glial cells at the lesioned site, especially at the interface between the gray and white matter, was clearly observed (*Figure 2C and D*).

We examined the cellular and neurotransmitter composition of the AMY and PAG, as these structures are downstream from the ACC and their natural development may be affected by the infant ACC lesions. We first investigated whether neurons in these structures were degenerated using Fluoro-Jade C (Histo-Chem Inc), which is used as a marker for apoptotic, necrotic, and autophagic cells. There was no sign of neurodegeneration in these downstream brain regions 2 years following the infant lesion. We next examined the proportion of neurons in the AMY and PAG expressing GABA, since changes in the relative number of inhibitory interneurons could serve as a marker for downstream neuroplasticity in response to the ACC lesion (*Atapour et al., 2021*). We found a significant reduction of GABA positive neurons in two structures, the basomedial AMY and the dorsal portion of the PAG (*Figure 2E*). This reduction suggests a disruption of the normal inhibitory balance within the vocal network following the infant ACC lesions.

## Vocal behavior persists immediately following neonatal ACC lesions

In the weeks following bilateral ACC lesions, infants tested in the isolated chamber continued to vocalize readily, which is somewhat surprising given the critical role of this structure in normal vocal behavior (*Jürgens and Ploog, 1970*; *Müller-Preuss et al., 1980*; *Jürgens, 1976*). From (presurgical) postnatal week 2 to (postsurgical) postnatal week 6, we annotated 23,000 calls from the five lesioned and five control marmosets. Sample spectrograms from audio recordings of a twin pair before and after surgery are shown in *Figure 3A and B*. While there was variability among individuals, the calls were complex and diversified from postnatal week 2, consisting of cries, as well as immature versions of adult vocalizations, including phee, twitter, and trills, as well as complex calls when two calls merged such as trill-twitter or cry-phee, or any other combination (*Figure 3A*). By the sixth postnatal week, the call repertoire for both the lesioned animals and the controls had both evolved, with no conspicuous difference between groups (*Figure 3B*). Most notably, the relative reduction of the total rate and diversity of calls was similar between groups (*Figure 3C*, $\chi^2(2)=2.8464$, $p=0.24$), with the reduction matching the known maturational changes accompanying growth of the vocal apparatus and increased respiratory powers (*Zhang and Ghazanfar, 2018*). Further analysis, on the proportion of calls emitted following surgery, showed that the ACC lesion had minimal effects on the rate of most

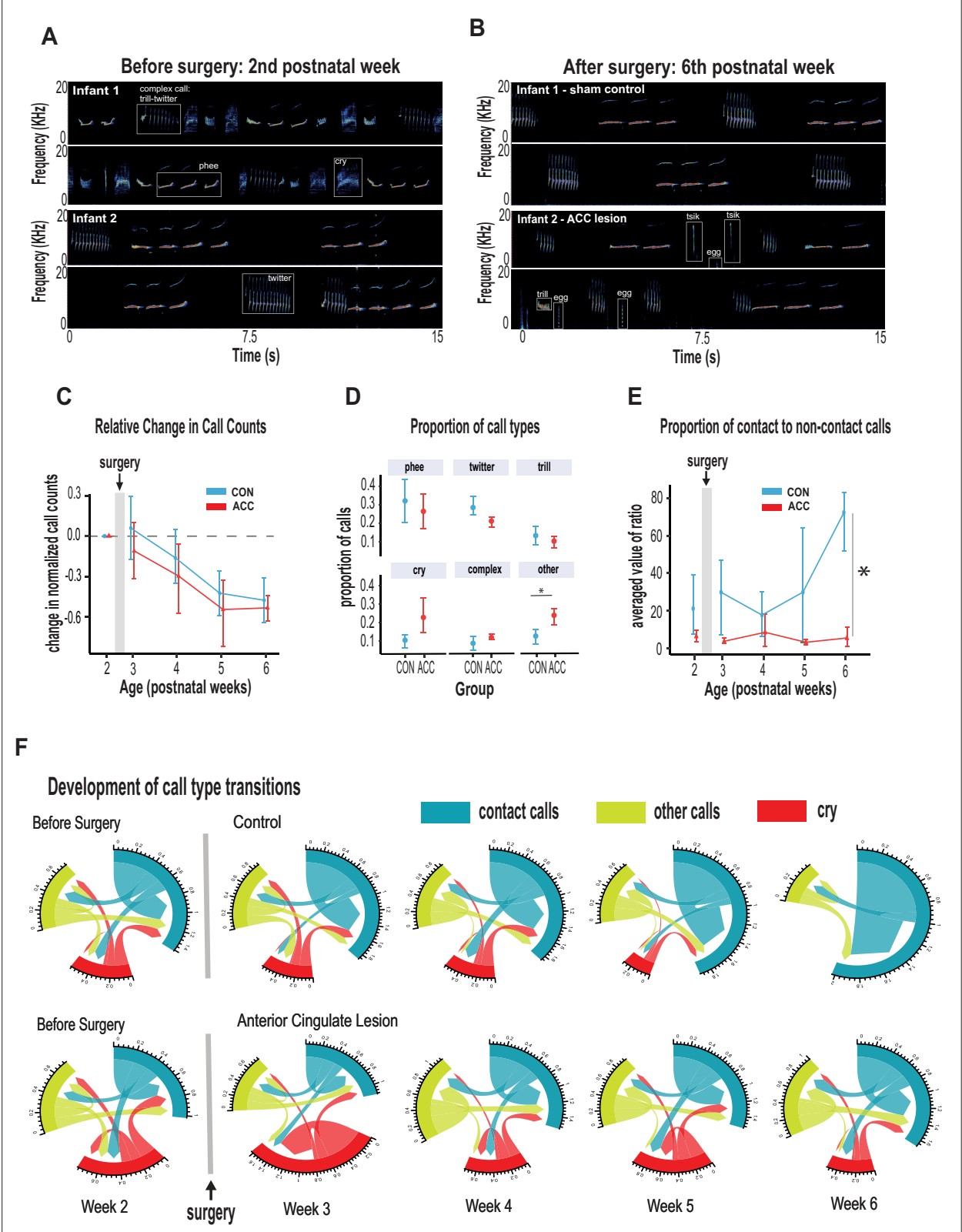

**Figure 3.** Anterior cingulate cortex (ACC) in early life is integral to postnatal development of social contact calls. (**A–B**) Spectrograms show sample 30 s vocal recordings of a representative control and ACC-lesioned marmosets before (postnatal week 2) and after surgery (postnatal week 6). Before surgery, the infants 'babbled' by emitting a wide range of immature concatenated calls, each with its own spectrogram motif illustrated and labeled in boxes. After surgery, at postnatal week 6, calls show reduced variability separated by distinct gaps or inter-call intervals. (**C**) Both groups show a reduction in the

*Figure 3 continued on next page*

*Figure 3 continued*

relative call count with increasing age. (**D**) Average proportion of each call type pooled from week 3 to week 6 following surgery. Animals in both groups were able to emit calls of different call types. Those with ACC lesions made minor calls designated as 'other' more frequently than controls, but all major call types were produced at equivalent rates. (**E**) Proportion of mature contact calls relative to immature non-social contact calls. The y-axis represents the averaged value of the ratios of the number of social calls divided by the number of nonsocial calls: $\bar{x}$ (# mature calls/# immature calls). Despite their ability to produce all call types, the proportion of mature contact calls comprising phee, twitter, and trills was substantially reduced in animals with early life ACC lesions at postnatal week 6. Due to factors beyond our control (COVID-19), the number of recordings varied between animals: week 3: CON n=5, ACC n=5; week 4: CON n=5, ACC n=4; week 5: CON n=4, ACC n=3; week 6: CON n=4, ACC n=3. (**F**) Chord diagrams illustrate the likelihood of transitioning between call types. At 6 weeks of age, animals with ACC lesions showed a higher likelihood of transitioning between all call types, but less frequent transitions between social contact calls relative to the sham group. The chord diagrams visualize the weighted probabilities and directionality of these transitions between different call types. Weighted probabilities were used to account for variations in call counts. The thickness of the arrows or links indicates the probability of a call transition, and the numbers surrounding each chord diagram represent the relative probability value for each specific transition.

The online version of this article includes the following figure supplement(s) for figure 3:

**Figure supplement 1.** Physical factors in developing marmosets.

call types during this period: phee (β=–0.07, 95% CI [–0.31, 0.17], p=0.49); twitter (β=–0.07, 95% CI [–0.16, 0.01], p=0.09); trill (β=–0.03, 95% CI [–0.11, 0.06], p=0.49); cry (β=0.13, 95% CI [–0.03, 0.29], p=0.10) (*Figure 3D*). The exception to this rule was an elevation in the rate of 'other' calls, which comprised tsik, egg, ock, chatter, and seep calls. These calls were significantly elevated in animals after the ACC lesion (β=0.11, 95% CI [0.03, 0.20], p=0.018). This was driven mostly by an increase in the lesion group during postnatal week 4.

Two additional variables relatively unaffected by the ACC lesion were the call durations and inter-call intervals, acoustic features that have been used to track vocal development in previous studies (*Takahashi et al., 2015*; *Gultekin and Hage, 2018*; *Gultekin et al., 2021*). Consistent with the overall decrease in vocalization rate with increasing age, there was an associated increase in inter-call intervals which was noted at late postnatal weeks (β=0.33, 95% CI [0.22, 0.43], p<0.001) which held true for both controls and lesioned group ($\chi^2$(2)=1.88, p=0.39). None of the specific call types exhibited developmental changes in call type duration phee ($\chi^2$(2)=1.08, p=0.58); trill ($\chi^2$(2)=2.87, p=0.24); twitter ($\chi^2$(2)=2.79, p=0.25); cry ($\chi^2$(2)=0.057, p=0.97), but there were slight changes in the duration of phee syllables exhibited only by animals with an ACC lesion, which we discuss later. Importantly, we did not observe ACC-lesion induced changes in physical growth factors such as body weight and grip strength which could feasibly impact vocal parameters such as duration (*Takahashi et al., 2015*; *Figure 3—figure supplement 1*).

## Neonatal ACC lesions prevent early maturation of social contact calls

Whereas many of the basic vocal parameters evolved normally in the animals with the ACC lesions, one major difference related to their use of social contact calls. By 6 weeks of age, marmoset vocalizations are known to approach their mature state and become dominated by social contact calls, namely phees, trills, and twitters. The phee call is studied most extensively as a long-distance contact call. It is typically evoked when the animal is socially distanced or isolated, and it promotes vocal exchanges between marmosets located out of sight in far-away locations (*Ghazanfar et al., 2001*; *Miller and Wang, 2006*) to facilitate reunion with family groups (*Bakker and Langermans, 2018*). Trills and twitters are short-distance contact calls thought to monitor the presence of group members (*Bezerra and Souto, 2008*; *Liao et al., 2018*). Since the ACC plays a major role in socioemotional cognition (for review, see *Devinsky et al., 1995*), we surmised that the ACC lesion might specifically influence the socioaffective content of the vocalization that is normally expressed through contact calls.

We thus grouped phee, twitter, and trill calls as mature contact calls and compared them with noncontact calls, namely tsik, egg, ocks, chatter, and seep. At 6 weeks of age, mature contact calls predominated the control animal's vocalization. However, in ACC-lesioned animals, this aspect of social vocal behavior was substantially reduced. This difference emerged gradually after the surgery and was only evident at 6 weeks of age (*Figure 3E*; $\chi^2$(2)=12.73, p=0.026). By the sixth week, the social vocal repertoire of the lesioned animals was altogether different from the control animals, with a much smaller proportion of social contact calls. We examined cries separately as immature social contact calls since parents are generally more responsive to infant cries, and this socially reinforces

vocal development (*Gultekin and Hage, 2017*; *Gultekin and Hage, 2018*). We found, however, that infant cry rates reduced substantially over the course of 6 weeks. In fact, the controls stopped crying at postnatal week 5 (percent cries compared to all call types: controls, week 5~3%, week 6~0%; ACC, week 5~20%, week 6~8.4%). This might be related to how cry-calls transition from immature to adult-like calls during babbling, thought to be accelerated with parental feedback (*Gultekin and Hage, 2018*).

To further understand the effect of the ACC lesion on the normal distribution of calls, we investigated the call transition probabilities between contact calls, cries, and other calls (*Figure 3F*). In contrast to the control animals, whose repertoire was dominated by social contact calls, the ACC lesion group showed frequent transitions mostly to other non-contact call types (u-index Wilcoxon test, p=0.055). These data suggest, therefore, that the ACC mediates developmental changes within the first 6 weeks of life that lead to the dominant production of isolation-induced contact calls and the gradual reduction of cries and other calls.

## Neonatal ACC lesions alter sequential characteristics of social contact calls

We examined the characteristics of vocal sequences to learn more about how early life ACC lesions might influence the acoustic signals that marmosets potentially relay to distantly located family members or other conspecifics when socially isolated. We focused on phee syllables, which are discrete elements or components of a call separated by very short intervals (*Devinsky et al., 1995*; *Agamaite et al., 2015*; *Figure 4A*). Thus, a sequence of phee calls may comprise multiple syllables (*Figure 4A*). The functional significance of syllables is not clearly understood, but a change in the number of syllables or their acoustic characteristics might feasibly alter the message conveyed to a family that cannot be seen or heard. This is especially important to young infant monkeys that are naturally demanding of attention, and even more so if isolated. The number of syllables per phee call was highly variable among the animals, ranging from 1 to 8 syllables per phee.

The ACC lesion did not greatly affect the phee syllable count. Aside from a transient decline in the number of syllables in the week after the surgery (postnatal week 3: Wilcoxon test p=0.042), these animals showed the normal preferred range of 3–4 phee syllables at later postnatal weeks (*Figure 4B*). Even in the sixth postnatal week, when the proportion of phee and other contact calls was much lower in the ACC-lesioned animals, the number of syllables in those phee calls that were issued was similar to the control group.

However, other phee call variables were affected by the lesion. For example, the duration of phee syllables was shortened in ACC-lesioned animals ($\chi^2$(5)=13.27, p=0.021), particularly in the later syllables of a multisyllabic phee. This effect emerged gradually and was most pronounced when the animals were 5–6 postnatal weeks (*Figure 4C*). Likewise, the amplitude of phee syllables was also affected by the ACC lesion ($\chi^2$(5)=48.178, p<0.0001) with lesioned animals making louder phee calls (*Figure 4D*). For each phee syllable, the amplitude difference between groups increased until postnatal week 5 and then disappeared at postnatal week 6 (postnatal week 4: β=3017.75, 95% CI [1831.61, 4203.89] p<0.001; postnatal week 5: β=3719.00, 95% CI [2389.88, 5048.12], p<0.001). In line with the increase in amplitude, the peak frequency of each phee syllable was also lowered by the ACC lesion (*Figure 4E*). This change occurred soon after the lesion (postnatal week 3: β=–488.63, 95% CI [–702.40, –274.85] p<0.001; postnatal week 4: β=–384.61, 95% CI [–607.98, –161.24], t(513) = –3.38, p<0.001).

Finally, we examined the entropy of phee syllables as a measure of vocal complexity (*Kershenbaum, 2013*). High entropy in multisyllabic phees would indicate that these vocalizations are diverse, variable, and unpredictable. We found that animals with ACC lesions exhibited lower entropy in phee syllables relative to controls as early as postnatal week 3 ($\chi^2$(5)=34.528, p<0.0001) (*Figure 4E*), suggesting that their vocal sequences were simple, less diverse, and stereotyped. Together, our data suggest that the ACC lesion compromised the normal development of the phee signature for each monkey by making them shorter, louder, and monotonic.

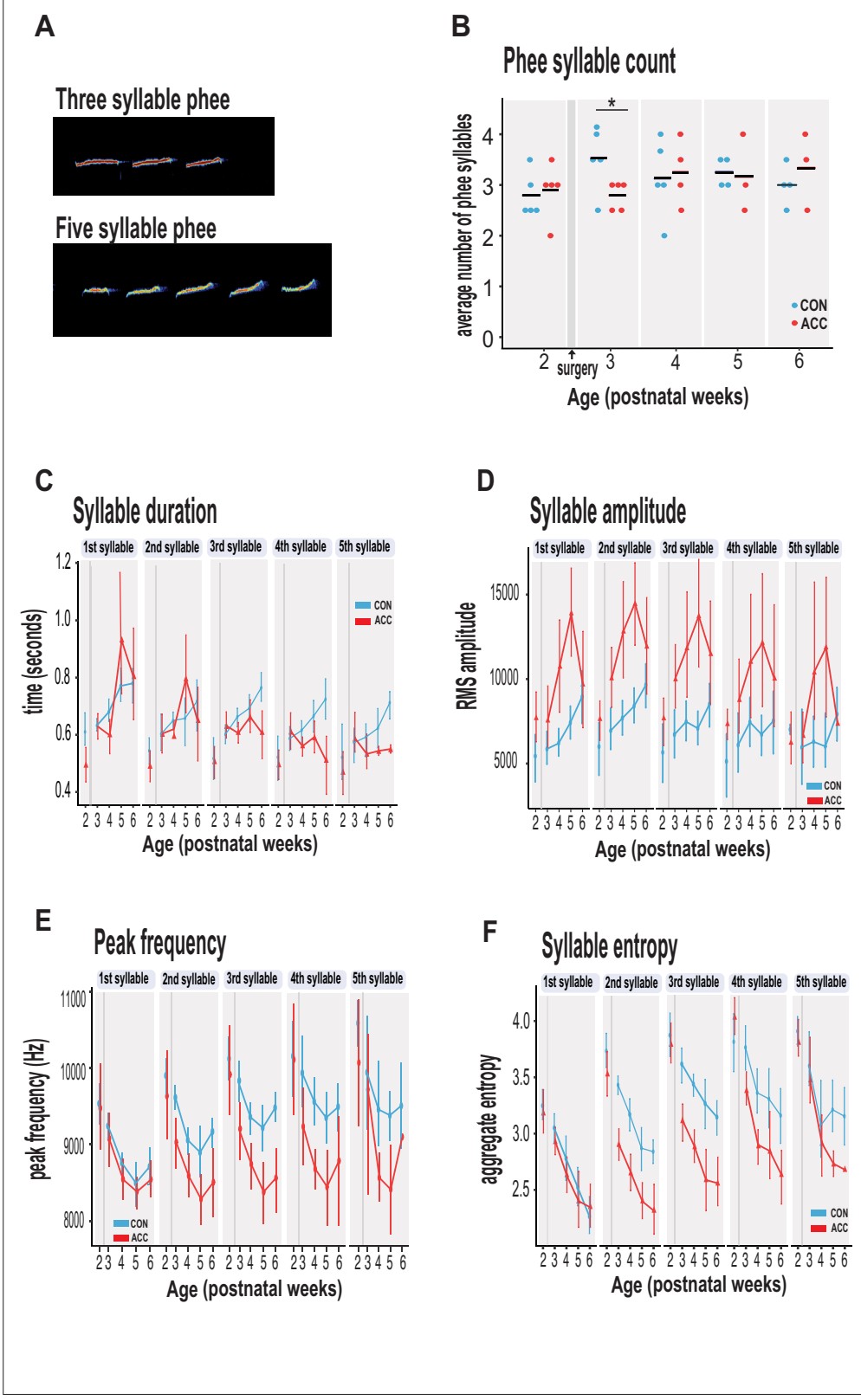

**Figure 4.** Anterior cingulate cortex (ACC) lesion alters structural characteristics of long-distance social phee calls. (**A**) Sample spectrogram with examples of three and five syllable phee calls. (**B**) The ACC lesion caused a reduction in average phee syllable counts immediately after the ACC lesion (red dots) at postnatal week 3, but then normalized to 3–4 syllables thereafter. (**C**) Phee syllable duration in the ACC-lesioned group became shorter for

*Figure 4 continued on next page*

*Figure 4 continued*

multisyllabic phees ≥3, especially with increasing age. (**D**) The effective amplitude for each phee syllable increased for the ACC group until postnatal week 6. (**E**) Animals with ACC lesions emitted low entropy phees for calls as low as 2 syllables and continued until postnatal week 6. (**F**) Decrease in peak frequency of phee calls immediately following ACC lesion. Data in D–F represents average of specific syllable in a phee sequence irrespective of the number of syllables in a phee (e.g. 1 syllable phee, the 1st syllable in a 2-syllable phee, in a 3-syllable phee, and in a 4-syllable phee). Due to factors beyond our control (COVID-19), the number of recordings varied between animals: week 3: CON n=5, ACC n=5; week 4: CON n=5, ACC n=4; week 5: CON n=4, ACC n=3; week 6: CON n=4, ACC n=3. Error bars are confidence intervals. Gray shaded lines or bars represent time of surgery.

## Discussion

We found that early life ACC lesions led to specific alterations in the production of vocal calls, developmental changes to the quality of social contact calls, and their associated sequential and acoustic characteristics were compromised. Contact calls that normally dominate the marmoset vocal repertoire at around 6 weeks were selectively diminished in the lesioned animals. When one common contact call, the phee call, was issued, its structural characteristics were unusual. The ACC lesions also led to permanent changes in remote brain areas known to be involved in vocal behavior. Notably, the proportion of presumptive inhibitory interneurons was reduced in both the basomedial AMY and dorsal PAG. We can infer from these findings that an intact ACC in early life is integral to postnatal development of social vocalizations, and that its interactions with vocalization-eliciting sites from a very early age are fundamental to the normal vocal expression of social behavior.

Consistent with previous reports, the range of call types observed in both neonatal controls and neonatal ACC-lesioned animals within the first weeks of postnatal development was stereotyped and repetitive (*Gultekin et al., 2021*; *Elowson et al., 1998*). With increasing age, the call rate gradually declined such that by the time the animals were 6 weeks of age, the most common vocalizations were those that conveyed social distance. Such calls solicit attention from family members and may trigger a range of behaviors including search, approach, interaction, and caregiving in response to the need for social contact. This is especially so for infants whose well-being depends on social feedback and reciprocal interaction (*Takahashi et al., 2015*; *Gultekin and Hage, 2017*; *Miller and Wang, 2006*; *Takahashi et al., 2017*). Our data suggest that the ability to effectively convey this social need was significantly altered in animals with early life ACC damage. At 6 weeks of age, these animals were not making social vocalizations at the same high rate as their age-matched controls. This reduction in social vocalizations does not appear to reflect a general slowing in vocal maturation, since other call types had advanced at the normal rate.

Since ACC lesions in humans cause social apathy (*Eslinger and Damasio, 1985*, *Damasio et al., 1990*), one possibility is that early life ACC removal altered the animals' desire or motivation for social reinforcement; these infants appeared to make little effort in using vocalizations to solicit social contact when socially isolated. This change in call usage aligns with their social development period at around postnatal week 6 when infant marmosets transition from using fixed, stereotypical calls to a flexible and more individualized call repertoire as they wean toward independence (*Gultekin et al., 2021*). Our data suggest that this transition does not occur normally following an ACC lesion.

In addition, the neonatal ACC lesion altered the quality of the infants' long-distance phee calls; they were shorter in duration, louder in amplitude, lower in peak frequency, and abnormal in their entropy such that the acoustics of calls were blunted of variation and less diverse. This suggests that the social message conveyed by these infants to their families through phee calls, even though it was loud and propagating over long distances, was potentially deficient, limited, and/or indiscriminate. However, the impact of entropy on emotional quality of vocalizations has not been systematically explored. Generally speaking, high entropy relates to high randomness and distortion in a signal. Accordingly, one view posits low-entropy phee calls represent mature sounding calls relative to noisy and immature high-entropy calls (*Takahashi et al., 2017*). In the current study, the reduction in syllable entropy observed for both groups of animals with increasing age is consistent with this view.

At the same time, entropy relates to vocal complexity; high entropy refers to complex and variable sound patterns, whereas low entropy sounds are predictable, less diverse, and simple vocal sequences (*Kershenbaum, 2013*). One possibility is that call maturity does not equate directly to emotional quality. In other words, a low-entropy mature call can also be lacking in emotion as observed in

humans with ACC damage; these patients show mature speech, but they lack the variations in rhythms, patterns, and intonation (i.e. prosody) that would normally convey emotional salience and meaning. Our observation of a reduction in phee syllable entropy in the ACC group in the context of being short, loud with reduced peak frequency is consistent with this view and suggests that the ACC group was emitting phee calls that were potentially lacking emotional meaning.

The long-term behavioral implications of such imperfect vocalizations are currently unknown but could, ostensibly, affect their ability to use long-distance social vocalizations to maintain intragroup functions such as warn of predators, strengthen family bonds, and maintain group cohesion more generally (*Oliveira and Ades, 2004*). Since the ACC exerts regulation over autonomic responses (*Robinson and Mishkin, 1968*; *Neafsey, 1990*; *Buchanan and Powell, 1993*), its ablation so early in life might feasibly blunt respiratory and vocalization responses in negative emotional environments such as social isolation. How these factors impact vocal behavior is a current topic of investigation (Sheikhbahaei et al., *SfN* abstracts, 2023).

Although we found that an intact ACC in early life is integral to the postnatal maturation of social vocalizations, we also show that it is not critical for production of infant vocalizations more generally. A number of early observations reporting the loss of learned or spontaneous vocalizations following bilateral ACC lesions left this question open, though it has been clear that vocal production in adults can withstand ACC damage (*Sutton et al., 1974*; *MacLean and Newman, 1988*; *Trachy et al., 1981*; *von Cramon and Jürgens, 1983*; *Jürgens, 1998*). In addition to ablating the ACC, researchers found that it was necessary to ablate other frontolimbic areas to permanently eliminate infant vocalizations such as cries (*MacLean and Newman, 1988*; *MacLean, 1985*). Our findings indicate that innate vocal production in the earliest phases of life, as early as 2 weeks postnatally in the marmoset, is not critically dependent on the ACC. The infant 'babbling' behavior observed in marmosets and other primates (*Takahashi et al., 2015*; *Elowson et al., 1998*; *Pistorio et al., 2006*) was largely preserved in the ACC-lesioned infant monkeys which, like the control group, produced long sequences of concatenated calls composed of rudimentary features of mature adult-like calls.

While the ACC is not essential for infant vocal behavior, its absence affects not only the maturation of social vocal behavior but also the anatomical compositions of structures with which it is interconnected. We noted a decrease in presumptive inhibitory interneurons in the dorsal PAG and basomedial AMY, two prominent ACC target regions involved in vocal behavior (*Jürgens, 2002*; *Vogt and Barbas, 1988*; *Jürgens, 2009*). We can speculate that this reduction might stem from a prolonged deafferentation of cingulate inputs, gradually leading to a rebalancing of the excitatory/inhibitory elements in the local circuit.

One potentially related observation is that phee calls became louder in the weeks following the surgery. It is interesting to speculate that such amplitude increases might reflect a local decrease in inhibition in structures such as the AMY or PAG, whose activity is thought to tune the emotional characteristics of social vocalizations. The primate ACC receives dense projections from the basomedial AMY with notably minimal direct input from the lateral nucleus (*Amaral and Price, 1984*; *Carmichael and Price, 1996*; *Aggleton et al., 2015*), and layer V pyramidal neurons project to the PAG (*Mantyh, 1982*; *Morrell et al., 1981*) with a greater concentration directed to the dorsolateral column (*Bandler, 1988*). Thus, the ACC has the capacity to directly activate the brainstem vocalization pathway as early as the first few weeks of life. Both the AMY and PAG are highly active during contexts in which threat-related vocalizations would normally be triggered (*Fecteau et al., 2007*; *Mobbs et al., 2007*; *Reis et al., 2021*), and both regions elicit vocalizations through electrical stimulation or pharmacological disinhibition (*Jürgens and Ploog, 1970*; *Robinson, 1967*; *Jürgens and Lu, 1993*; *Forcelli et al., 2017*). We cannot be sure when during postnatal development the ACC lesion altered GABA expression in the AMY and PAG, but from our results, we can infer that the appropriate modulation and coordination of social vocal behavior requires the normal postnatal development of the ACC.

Existing evidence in monkeys and humans demonstrates unequivocally the importance of the ACC in its contribution to emotional vocalization. In humans, as in monkeys, ACC lesions do not eliminate vocal behavior, but instead tend to remove the intonation and prosodic features of the vocalization characterized as expressionless (*von Cramon and Jürgens, 1983*). This is consistent with the changes observed in the marmoset phee calls. In general, our data suggest that the ACC shapes the emotional structuring of social calls during the first few weeks of life in the marmoset. Given the many similarities to humans, and the strong contribution of socioemotional information to the vocal productions

beginning in infancy, it is reasonable to speculate that similar mechanisms apply to the development of early life human vocal behavior. Importantly, our data confirm the importance of vocalizations as a means of conveying social information even when family members or conspecifics are not physically present. Our data suggest that in the absence of a functioning ACC in early life, infant calls conveying social information that would elicit feedback from parents and other family members may be compromised, and this could potentially influence how that infant develops its social interactive skills. The ability to normalize brain circuits in early life would provide a major therapeutic advance for the remedial treatment of social deficits that plague disorders of mental health.

## Materials and methods
### Subjects
All procedures accorded strictly with the Guide for the Care and Use of Laboratory Animals and were approved by the Animal Care and Use Committee of the National Institute of Mental Health (#SBN-02). A total of 10 marmosets (Callithrix jacchus), five males and five females, all born in captivity, were used in this study. Five infants received ACC lesions at 14–16 postnatal days of age. Five others served as age-matched controls. The infants were raised by parents and siblings in family groups comprising four to six members and housed in temperature-controlled rooms (~27°C), 50–60% relative humidity under diurnal conditions (12 hr light:12 hr dark). Food and water were available ad libitum, supplemented with fresh fruit or vegetables. One animal showed sparing of the ACC lesion in one hemisphere. The final sample size for the behavioral data was n=5/per group. For the MRI and histological data, the final sample size was n=4/per group.

### Surgery
We first obtained a reference MRI scan using a 14-day-old ex vivo sample (*Figure 1A*). A T2-weighted scan was obtained using 7T Bruker Biospin MRI platform with an eight-channel volume coil. Using ParaVision Acquisition 6.0.1, the following echo sequence was used to acquire a three-dimensional volume of the infant marmoset brain: TR = 400, TE = 72 ms, flip angle = 90 degrees, matrix size = 256 × 256 × 214, resolution = 0.15 mm isotropic, number of averages = 8, number of repetitions = 1, and the total scan time was 3 hr. The scan was aligned horizontally by rotating the image until the anterior and posterior commissures were positioned at the same height and water-filled ear bars were used to obtain the interaural reference. We then used ITK-SNAP (*Yushkevich et al., 2006*) to identify the ACC at the coronal planes before the genu of the corpus callosum to the level of the anterior commissure. The coordinates were calculated relative to the ear bars and midline references, both of which were visible on the scan. The resulting 14-day-old marmoset scan served as a template atlas to calculate injection coordinates to target the rostral portion of the dorsal ACC (24a and 24b), bilaterally, in all marmosets. We calculated five injection coordinates for each hemisphere: (1) AP: 10.7 mm, ML: ±0.7 mm, DV: –2.9 mm; (2) AP: 10.7 mm, ML: ±0.7 mm, DV: –4.5 mm; (3) AP: 9.5 mm, ML: ±0.7 mm, DV: –3.5 mm; (4) AP: 8.5 mm, ML: ±0.7 mm, DV: –3.6 mm; (5) AP: 7.5 mm, ML: ±0.7 mm, DV: –3.5 mm.

The entire surgical procedure was performed under aseptic conditions in infant monkeys that were 14–16 days of age. During surgery, monkeys received isotonic fluids. Heart and respiration rates, body temperature, blood pressure, and expired $CO_2$ were monitored throughout the procedure. Pre- and postoperatively, monkeys received nonsteroidal anti-inflammatory drugs (meloxicam, 2 mg/ml, s.c.) to reduce swelling. The monkey was first immobilized with an anesthetic dose of alfaxalone (10 mg/kg, i.m.) combined with diazepam (5 mg/kg, i.m.). In this state, the infant's head was shaved, and vital electrodes were secured on the infant's chest. Temperature was measured with a rectal probe. The infant's head was then secured in a small animal stereotaxic frame (Stoelting Company, IL, USA) attached to a custom-built stage fitted with eye bars, ear bars, and a pallet bar to accommodate the small head. Once the head was secured in the frame, anesthesia with isoflurane gas (1–2% to effect) was provided through the custom-fitted mask (*Figure 1B*). An integral part of the pallet bar was a gas hole (0.6 mm diameter) that ran along the length of the bar and connected, via tubing, to a vital monitor to measure small end-tidal $CO_2$ volumes.

Following a midsagittal incision, the scalp was retracted, and a craniotomy was made above the target coordinates of the brain. A 5 µl syringe (33 gauge, Neurosyringe, Hamilton Company, Reno, NV, USA) was used to administer bilateral injections of 0.12 M NMDA (M3262, Sigma-Aldrich) dissolved

in sterile-filtered saline into the ACC (0.5 µl per injection site). Each injection was made over 2 min and the injector remained in place for an additional 4 min for dispersion. When all injections were complete, the scalp was closed with intradermal absorbable sutures, and the infant was allowed to recover in an intensive care unit that was void of extraneous sensory stimulation (e.g. excessive bright lights and loud noise). During recovery, marmosets received a combination of Esbalic and Enfamil (3:1 ratio) infant formula every 2–3 hr. When fully awake, each infant was returned to its family unit. A total of five marmosets received the neonatal cingulate lesion. Another five marmosets served as controls: two received saline injections (shams), one received a craniotomy only, and another two were unoperated.

## Vocal recordings

Each infant was placed in a temperature-controlled incubator set to ~38°C (ThermoCare, CA, USA), and the emitted vocalization was recorded for 5 min. Sound recordings were acquired using a cardioid microphone (Sennheiser ME 64, Sennheiser, Wedemark, Hanover, Germany) that was placed on the side of the incubator (*Figure 1C*). The microphone was connected to a computer, and recorded sounds were digitized at a sampling frequency of 44 kHz using Raven Lite software (Cornell Lab of Ornithology, Ithaca, NY, USA). Due to a variety of extraneous factors beyond our control, including restrictions due to the COVID-19 pandemic, the exact day and number of recordings differed between monkeys. Therefore, recording sessions from each infant were grouped by week. All recording sessions were conducted without the presence of investigators in the recording room. The infant was then returned to its family unit.

## Acoustic analysis

The spectrogram of each audio file was obtained and visually inspected using Raven Pro 1.6 (Cornell Lab of Ornithology, Ithaca, NY, USA). Spectrograms were generated with a Hann window of 512-sample points to filter the signal at 3 dB bandwidth of 124 Hz (e.g. *Figure 2A and B*). The calls were manually classified by a defined classification system (*Bezerra and Souto, 2008*; *Pistorio et al., 2006*). To identify call types in the spectrogram of a recording, Raven software features, such as amplitude waveform, spectrogram, and audio playbacks, were used. Six major call types (phee, trill, trill-phee, twitter, cry, and complex calls) were identified from spectrograms, along with other minor call types (tsiks, chatter, egg, ock, seep). In some cases, when trill-phees and phees looked similar in spectrograms of a recording, acoustic parameters such as entropy were used to carefully classify calls. Complex calls comprised of vocalizations with elements from at least two different simple call types such as trill-twitter or twitter-phee, etc. From each recording, call types were manually annotated by three trained investigators (inter-rater reliability >80%). The spectrograms were used to obtain acoustic measurements such as peak frequency, RMS amplitude, and aggregated entropy and exported for further analysis.

Call structures such as inter-call intervals, syllable duration, and number of phee syllables were analyzed using a custom-written R script (Nagarajan G, https://github.com/Guru-learn/Cingulo-tomy_Vocalization; copy archived at *Guru-learn, 2025*; *Nagarajan, 2024*). Acoustic analyses were performed only for phees, which served as the major call type because of their abundance during postnatal weeks. In some cases, vocalization quantity and amplitude were largely suppressed for several minutes after handling by the experimenters. Consequently, analysis of each recording began after 2 min had elapsed.

## Lesion assessment with MRI

Gross ACC volume was measured from anesthetized MR scans performed at approximately 8 months of age. Anesthesia was induced with 5% isoflurane. The animals were then placed in an MR-compatible cradle where their heads were secured using ear bars. Isoflurane was maintained at 1.5–2.5%, and vitals were monitored with V9004 Series Capnograph Monitor (San Clemente, CA, USA). We obtained T2-weighted scan (n=4 for control and n=4 for ACC) using the MR procedure described above with the following echo sequence: TR = 30, TE = 48, matrix size = 144 × 144 × 128, resolution = 0.25 mm isotropic, number of averages = 8, number of repetitions = 1. The rostro-caudal extent of the ACC was segmented and measured using ITK-SNAP 4.0. A representative sagittal view of the lesioned and

non-lesioned ACC can be seen in *Figure 1E*. Voxels containing ACC were carefully labeled from the anterior to posterior slice of the MR scan for each subject.

## Histological preparation and quantification

At approximately 24 months of age, the marmosets were euthanized and perfused with 0.1 M PBS followed by 4% paraformaldehyde. The brains were extracted and cryoprotected in 0.1 M phosphate-buffered sucrose (in steps of 10%, 20%, and 30% wt/vol). The brains were partitioned along the midline, and right hemispheres were used for further histological processing after sectioning in coronal orientation on a sliding microtome and cryostat into 40 μm sections.

## Immunohistochemistry

The immunohistochemistry (IHC) was performed on two series of free-floating sections. Initially, from each of the eight brains, six sections each were collected around three rostrocaudal planes at the following approximate locations (in reference to the interaural axis): +13.30 mm (target brain region: the ACC); +9.20 mm (target brain region: the AMY); and +2.05 mm (target brain region: the PAG matter). Subsequently, these 18 sections were divided into two IHC series of nine sections each, where three sections covered each target brain area for separate processing. The first IHC series was processed to visualize major cell classes that are likely to be affected by a brain lesion (neurons: primary antibody against NeuN, astrocytes: primary antibody against the glial fibrillary acidic protein [GFAP], microglia/macrophages: primary antibody against the ionized calcium-binding adaptor molecule 1 [Iba1]). The second IHC series was processed to visualize and assess the ratio of GABAergic neurons to all neurons: such changes in the relative number of inhibitory neurons could indicate local downstream neuroplasticity in reaction to the ACC lesion, as GABAergic neurons are primarily responsible for local inhibition. In the second IHC series, we visualized the distribution of NeuN, as well as of neurotransmitter GABA and GAD67, a rate-limiting enzyme in GABA synthesis that produces more than 90% of GABA in the central nervous system.

The first IHC series was incubated in a cocktail of the following primary antibodies for 60 hr at 4°C: NeuN (chicken, ABN91, Millipore Sigma), 1:500 dilution; GFAP (goat, SAB2500462, Millipore Sigma), 1:1500 dilution; Iba1 (rabbit, 019-19741, Fujifilm Wako Chemicals), 1:1500 dilution. Then, after several washes, the sections were incubated in a cocktail of the following secondary antibodies for 2 hr at RT: 1:200 donkey anti-chicken Alexa Fluor 488 (703-545-155, Jackson ImmunoResearch), 1:500 donkey anti-goat Alexa Fluor 594 (A21207, Invitrogen), 1:400 donkey anti-rabbit Alexa Fluor 680 (711-625-152, Jackson ImmunoResearch). Additional antibodies used in the second IHC series were as follows. Primary: GABA (rabbit, A2052, Millipore Sigma), 1:600 dilution; GAD67 (mouse, MAB5406, Millipore Sigma), 1:500 dilution. Secondary: 1:500 donkey anti-rabbit Alexa Fluor 594 (A21207, Invitrogen), 1:400 donkey anti-mouse Alexa Fluor 680 (A10038, Invitrogen). Finally, the sections were mounted on a glass slide, air-dried, and coverslipped with DEPEX mounting media (13515, Electron Microscopy Sciences, Hatfield, PA, USA).

## Myelin staining

Myelinated fibers were stained with aurohalophosphate-based Black-Gold II (Histo-Chem Inc, Jefferson, AR, USA) as shown in *Schmued et al., 2008*, according to the manufacturer's protocol and coverslipped with Permount (SP15, Fisher Scientific).

## Staining for degenerating neurons

Fluoro-Jade C (Histo-Chem Inc) stain was used to visualize degenerating neurons according to the manufacturer's protocol in the target brain regions of ACC, AMY, and PAG. There was no observable signal of neurodegeneration in the studied regions.

## Imaging and cell counting

Each histological section was digitized at 0.65 μm resolution (×10 magnification) using a Zeiss Axioscan microscope slide scanner. Images were then split into a separate channel for each fluorophore. Cell detection and counting were done with an open-source QuPath software 0.3.2 (*Bankhead et al., 2017*). As each fluorescence channel was analyzed separately, the loci of immunofluorescence that were counted do not necessarily correspond to unique cells, especially for microglia and GABA

channels where the fluorescent signal was more diffuse in appearance. Cell segmentation in brain regions downstream to the lesion site was done using the random trees algorithm embedded into QuPath. Cell segmentation of the lesion site was done by custom-trained Cellpose models (doi: 10.1038/s41592-022-01663-4). Although the researcher performing cell counts was blinded to the marmosets' identity, it was possible to identify the site of the lesion and determine the animals' group membership. Raw cell counts were transformed into cell densities per mm2 to account for size differences in region of interest (ROI) areas.

## Statistical analysis

All analyses were performed using R 4.2.3 (https://www.R-project.org/). For vocalization analysis, recording sessions were averaged by week, and unless otherwise noted, the data are represented as the mean values with mean ± confidence interval. Recording sessions obtained before the surgery were grouped as postnatal week 2 (pre-surgery), and all the recordings after the surgery were binned into postnatal weeks 3–6 (post-surgery). Due to COVID-19-related restrictions, recordings for some infants could not be extended beyond postnatal week 3.

Packages in the tidyverse library (*Wickham et al., 2019*) were used for data processing and analyses. Linear mixed effect models (LMMs) were used to analyze postnatal datasets, and these models were fitted with the `lmer()` function in lme4 package (*Bates et al., 2015*). For the estimation of coefficients, the maximum likelihood method was used. Models were fitted with postnatal weeks and experimental groups as fixed factors, including an interaction term between treatment and postnatal weeks to assess whether the effect of weeks differed across treatment groups. To account for inter-individual variability, each monkey was modeled as a random effect. For multi-syllabic phee analysis, syllables were nested within monkeys' random effect. Models with and without lesions were used to test the effect of the ACC lesion, and the lesion effect was considered significant at an α of 0.05 (*Winter, 2013*; *Brown, 2021*). Model assumptions were tested using the `check_model()` function available in the performance package. Log transformation was performed on some dataset to meet LMM assumptions. When the normality assumption was violated, a non-parametric test (Wilcoxon test) was also used. Graphs were created using the package ggplot2.

For analysis involving immunofluorescence, there were some inhomogeneities in the spatial distribution of IHC signal. The raw immunopositive detections for specific markers were normalized by NeuN cell counts that were obtained in the same QuPath processing pipeline. The final measure for each antigen-specific immunopositivity count was the ratio computed from the number of antigen-positive detections divided by the sum of the antigen-positive detections and NeuN-positive cells. This transformation bounded the possible antigen-specific detections between 0 and 1 and allowed for parametric modeling with beta distribution. Cell counts were fitted using the glmmTMB software package in R statistical computing environment (https://github.com/kissmyjazz/cingulotomy-histology, copy archived at *Matrov, 2025*). As there were three sections per animal for each ROI, these were modeled as random effects. Widths of major fiber tracts were modeled by a linear regression (function `lm` in base R). Final p-values were adjusted for multiple comparisons with the false discovery rate method for each antigen.

## Acknowledgements

This research was supported by the Intramural Research Program of the National Institute of Mental Health (ZIAMH002951 and ZICMH002952 to YC). The contributions of the authors were made as part of their official duties as National Institutes of Health (NIH) federal employees, are in compliance with agency policy requirements, and are considered Works of the United States Government. However, the findings and conclusions presented in this paper are those of the authors and do not necessarily reflect the views of the NIH or the U.S. Department of Health and Human Services. We thank George Dold, William Bennett, and David Ide from the NIMH Section on Instrumentation for customization of the stereotaxic frame and surgical anesthesia gas mask. We would also like to thank the Veterinary Medicine and Resources Branch and Central Animal Facility for animal husbandry, technical, and anesthetic support during procedures. GN is now at The Henry Jackson Foundation for the Advancement of Military Medicine, Bethesda, MD, USA. AP is now at Georgetown University, Washington DC, USA.

# Additional information

## Funding

| Funder | Grant reference number | Author |
|---|---|---|
| National Institutes of Health | Intramural Research Program ZIAMH002951 | Yogita Chudasama |
| National Institutes of Health | Intramural Research Program ZICMH002952 | Yogita Chudasama |

The funders had no role in study design, data collection and interpretation, or the decision to submit the work for publication.

## Author contributions

Gurueswar Nagarajan, Conceptualization, Software, Formal analysis, Investigation, Visualization, Methodology, Writing – original draft, Writing – review and editing; Denis Matrov, Software, Formal analysis, Investigation, Visualization, Writing – review and editing; Anna C Pearson, Investigation, Writing – review and editing; Cecil C Yen, Investigation; Sean P Bradley, Writing – review and editing; Yogita Chudasama, Conceptualization, Resources, Supervision, Funding acquisition, Investigation, Visualization, Methodology, Writing – original draft, Writing – review and editing

## Author ORCIDs

Gurueswar Nagarajan (ID) https://orcid.org/0000-0003-2661-6148
Denis Matrov (ID) https://orcid.org/0000-0002-3962-3008
Anna C Pearson (ID) https://orcid.org/0000-0001-9025-0078
Sean P Bradley (ID) https://orcid.org/0000-0002-7557-6560
Yogita Chudasama (ID) https://orcid.org/0000-0003-3349-8477

## Ethics

This study was performed in strict accordance with the recommendations in the Guide for the Care and Use of Laboratory Animals and were approved by the Animal Care and Use Committee of the National Institute of Mental Health (#SBN-02).

Reviewer #1 (Public review): https://doi.org/10.7554/eLife.97125.3.sa1
Reviewer #2 (Public review): https://doi.org/10.7554/eLife.97125.3.sa2
Reviewer #3 (Public review): https://doi.org/10.7554/eLife.97125.3.sa3
Author response https://doi.org/10.7554/eLife.97125.3.sa4

# Additional files

## Supplementary files

MDAR checklist

## Data availability

Source files for all figures and custom code for analysis can be found in Mendeley Data repository: https://data.mendeley.com/datasets/wf48t9b7t6/1. Additional code for figures is located in https://github.com/Guru-learn/Cingulotomy_Vocalization (copy archived at *Guru-learn, 2025*). Code for histological quantification can be found at https://github.com/kissmyjazz/cingulotomy-histology (copy archived at *Matrov, 2025*).

The following dataset was generated:

| Author(s) | Year | Dataset title | Dataset URL | Database and Identifier |
|---|---|---|---|---|
| Gurueswar N, Denis M, Anna CP, Cecil Y, Sean PB, Yogita C | 2025 | Cingulate cortex shapes early postnatal development of social vocalizations | https://data.mendeley.com/datasets/wf48t9b7t6/1 | Mendeley Data, 10.17632/wf48t9b7t6.1 |

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
