## [Editor Report · eLife Assessment]

This **important** study investigates the influence of the cingulate cortex on the development of the social vocalizations of marmoset monkeys by making bilateral lesions of this brain area in neonatal animals. The evidence supporting the authors' claims is **convincing**. The work will be of broad interest to cognitive neuroscientists, speech and language researchers, and primate neuroscientists.

---

## [Referee Report · Reviewer #1 (Public review)]

Summary:

This study seeks to quantify changes in vocal behavior during development in marmosets with bilateral anterior cingulate cortex (ACC) lesions. The ACC and its role in social vocal behaviors is of great interest given previous literature on its involvement in initiation of vocalizations, processing emotional content, and its connectivity to two other critical nodes in the vocal network, the amygdala and the PAG. The authors seek to test the hypothesis that the ACC contributes to the development of mature vocal behaviors during the first few weeks of life by disrupting this process with neonatal ACC lesions. Imaging and histological analyses confirm the extent of the lesion and suggest downstream effects in connected regions. Analysis of call rates and call type proportions show no or slight differences between lesioned and controlled animals. Additional analyses on the proportion of grouped 'social' calls and certain acoustic features of a particular call, the phee, reveal more distinct differences between the groups.

Strengths:

The authors have identified that ACC lesions in early life have no or little influence on certain aspects of vocal behavior (e.g. call rate, call intervals) but larger impacts on other aspects (e.g. acoustic features of phee calls). This is difficult data to collect, especially in the difficulties of that particular time period. This data is a valuable addition to the literature on the effects of the ACC on vocal production and sparks intriguing follow-up questions on the role of different acoustic features (as related to emotional content) on vocal interactions with conspecifics over the lifespan.

The histological methods and resulting quantification of neural changes in the lesioned area and in downstream areas of interest are intriguing given the large time gap between the lesion and these analyses.

The changes to the text, figures, and additional supplemental figures to my previous review requests have made it easier to determine if conclusions are supported by the data in the manuscript. Examples include the quantification of the loss of neurons and increase in glial cells, the inclusion of changes in body weight and grip strength that could also be a result from the lesions affecting vocal behavior, and additional details on analysis methods.

Weaknesses:

The article emphasizes vocal social behavior. However, marmoset infants are recorded in isolation which allows for examining the development of vocal behavior in that particular context - reaching out to conspecifics. The text now covers the relationship between 'social' information in calls and development in this particular context. However, early-life maturation of vocal behavior is strongly influenced by social interactions with conspecifics. For example, the transition of cries and subharmonic phees which are high-entropy calls to more low-entropy mature phees is affected by social reinforcement from the parents. And this effect extends cross-context, where differences in these interaction patterns extend to vocal behavior when the marmosets are alone. Together, the results are interesting and important but may not fully capture the changes resulting from direct social interactions.

Additionally, it is an intriguing finding that the infants' phee calls have acoustic differences being 'blunted of variation, less diverse and more regular'. Though the text about how the social message conveyed by these infants was 'deficient, limited, and/or indiscriminate' is now better explained with additional text from human studies, it is still an assumption that this would directly translate to marmoset communication. Thus, experiments directed at the responses of other marmosets to these calls would still be important.

---

## [Referee Report · Reviewer #2 (Public review)]

Summary:

Nagarajan et al. investigate the role of the anterior cingulate cortex (ACC) in vocal development of infant marmoset monkeys using lesions in this brain area. Many previous studies show that ACC plays an important role in volitional and emotion-driven vocal behavior in mammals. The experiments Nagarajan et al. performed strengthen the long-standing hypothesis that ACC influences the development of social-vocal behavior in non-human primates. Furthermore, their anatomical studies support the idea of cortical structures exerting cognitive control over subcortical networks for innate vocalization, and thus, enabling mammals to perform flexible social-vocal communication.

Strengths:

Many invasive behavioral studies in monkeys often use 2-3 animals. The authors used a sufficiently high number of animals for their experiments. This increases the power of their conclusions.

The study also investigates the impact of ACC lesions on downstream areas important for innate vocal production. This adds further evidence to the role of ACC on influencing these subcortical regions during vocal development and vocal behavior in general.

Weaknesses:

The study only provides data up to the 6th week after birth. Given the plasticity of the cortex, it would be interesting to see if these impairments in vocal behavior persist throughout adulthood or if the lesioned marmosets will recover their social-vocal behavior compared to the control animals. The authors give a reasonable explanation for why they did not provide this data.

Even though this study focuses entirely on the development of social vocalizations, providing data about altered social non-vocal behaviors that accompany ACC lesions is missing. This data can provide further insights and generate new hypothesis about the exact role of ACC in social-vocal development. For example, do these marmosets behave differently towards their conspecifics or family members and vice versa, and is this an alternate cause for the observed changes in social-vocal development? Unfortunately, the authors are unable to provide that data. Hopefully, this will be the goal of future studies.

---

## [Referee Report · Reviewer #3 (Public review)]

Summary:

In this manuscript, Nagarajan et al. study the impact of early damage to the anterior cingulate cortex (ACC) on the vocal development of marmoset monkeys. AAC lesions were performed on neonatal marmosets and their vocal patterns and the spectrotemporal features of their calls were analyzed compared to control groups during the first six weeks of life. While the vocal repertoire was not significantly affected by ACC lesions, the authors described notable differences in the social contact call, the phee call. Marmosets with ACC damage made fewer social contact calls, and when they did, these calls were shorter, louder, and monotonic. Additionally, the study revealed that ACC damage in infancy led to permanent alterations in downstream brain areas involved in social vocalizations, such as the amygdala and periaqueductal gray.

Strengths:

This study suggests that the ACC plays a crucial role in the normal development of social vocal behavior in infant marmosets. Studying vocal behavior in marmosets can provide insights into the neural mechanisms underlying human speech and communication disorders due to their similarity in brain structure and social behavior.

The methods are robust and reliable with precise localization of the lesions with neuroimaging and histological examination.

---

## [Author Response]

The following is the authors’ response to the original reviews

**Reviewer #1 (Public Review):**
The article emphasizes vocal social behavior but none of the experiments involve a social element. Marmosets are recorded in isolation which could be sufficient for examining the development of vocal behavior in that particular context. However, the early-life maturation of vocal behavior is strongly influenced by social interactions with conspecifics. For example, the transition of cries and subharmonic phees which are high-entropy calls to more low-entropy mature phees is affected by social reinforcement from the parents. And this effect extends cross context where differences in these interaction patterns extend to vocal behavior when the marmosets are alone. From the chord diagrams, cries still consist of a significant proportion of call types in lesioned animals. Additionally, though it is an intriguing finding that the infants' phee calls have acoustic differences being 'blunted of variation, less diverse and more regular,' the suggestion that the social message conveyed by these infants was 'deficient, limited, and/or indiscriminate' is not but can be tested with, for example, playback experiments.

We recognize that our definition of vocal social behavior is not within the normal realm of direct social interactions. We were particularly interested in marmoset vocalizations as a social signal, such as phees, cries and twitter, even when their family members or conspecifics are not visibly present. Generally speaking, in the laboratory, infant marmosets make few calls when in the presence of another conspecific, but when isolated they naturally make phee calls to reach out to their distantly located relatives. In this context, while we did not assess the animals interacting directly, we assessed what are normally referred to as ‘social contact calls,’ hence the term ‘social vocalizations.’ Playback recordings might provide potential evidence of antiphonal calling as a means of social interaction and might reveal the poor quality of the social message conveyed by the infant, but even here, the vocalizing marmoset would be calling to a non-visible conspecific. Thus, although our experiment lacked a direct social element, our data suggest that in the absence of a functioning ACC in early life, infant calls that convey social information, and which would elicit feedback from parents and other family members, may be compromised, and this could potentially influence how that infant develops its social interactive skills. We have now commented on the significance of social vocalizations in the introductory text (page 3) and discussion (page 15).

The manuscript would benefit from the addition of more details to be able to better determine if the conclusions are well supported by the data. Understanding that this is very difficult data to get, the number of marmosets and some variability in the collection of the data would allow for the plotting of each individual across figures. For example, in the behavioral figures, which is the marmoset that is in the behavioral data that has a sparing of the ACC lesion in one hemisphere? Certain figures, described below in the recommendations for the authors, could also do with additional description.

Thanks for these suggestions. We have plotted the individual animals in the relevant figures and addressed the comments and recommendations listed below.

**Reviewer #1 (Recommendations For The Authors):**
Given the number of marmosets, variability in the collected data, lesion extent, and different controls, I would like to see more plots with individuals indicated (perhaps with different symbols). More details could also be added for several plots.

Figure 2D (new) and 2E now have plots that represent the individual animals, each represented by a different symbol.

(Figure 2A) Since lesions are bilateral, could you also show the extent of the lesions on the other side for completeness?

Our intention was to process one hemisphere of each brain for Golgi staining to examine changes in cell morphology in the ACC and associated brain regions following the lesion. Unfortunately, the Golgi stain was unsuccessful. Consequently, we were unable to use the tissue to reconstruct the bilateral extent of the lesion. We did, however, first establish the bilateral nature of the lesion through coronal slices of the animals MRI scan before processing the intact hemisphere to confirm the bilateral extent of the lesion. The MRI scans (every 5th section) for each control and lesioned animal is compiled in a figure in the supplementary materials (Fig. S1). These scans show that the ACC-lesioned animals have bilateral lesions with one animal (ACC1) showing some sparing in one hemisphere, as we noted in the text. We have now made reference to this supplemental figure in the text (page 5).

(Figure 2B/C) In Figure 2B, control and ACC lesions are in the columns while right next to it in 2C, ACC lesion and control are in the rows. Could these figures be adjusted so that they are consistent?

We have now adjusted these figures and updated the figure legends accordingly.

(Figure 2C) Is there quantification of the 'loss of neurons and respective increase in glial cells at the lesioned site especially at the interface between gray and white matter'? There are multiple slices for each animal.

Thanks for suggesting this. We have now quantified these data which are presented as a new graph as Fig. 2D. These data revealed a significant loss of neurons (NeuN) in the ACC group as well as an increase in glial cells (GFAP and Iba1) relative to the controls. The figure legend and results have also been updated.

(Figure 2C) It is difficult for me to distinguish between white and purple - could you show color channels independently since images were split into separate channels for each fluorophore?

Fig. 2C has been revised to better visualize the neurons and glia at the gray and white matter interface. We found that grayscale images for each channel offered a better contrast than separating the channels for each fluorophore.

(Figure 2C/D) I like how there are individual dots here for the individual marmosets. Since there are four in each group, could they be represented throughout with symbols (with a key indicating the pair and also the control condition)? For example, were there changes in the histology for control animals that got saline injections as opposed to those that didn't get any surgery?

We have highlighted the individual animals with different symbols in the figures. Although some animals were twin pairs, it was not possible to have twins in all cases. Only two sets were twins. We have indicated the symbols that represent the twin pair in Fig. 2 as well as the MRI scans of the twin pairs in Fig. S1. There were no observed changes in histology for the sham animals relative to the other non-sham controls. The MRI scan for one sham CON2 shows herniated tissue in the right hemisphere which is a normal consequence of brain exposure caused by a craniotomy.

(Figure 3D-E) Here, individual data points could be informative especially given that some animals are missing data past the third week.

To prevent cluttering the figure with too many data points, we have added the sample size for each group in the figure legend (pages 33).

*(Figure 3D/F) What exactly is the period that goes into this analysis? In the text, 'Further analysis showed that the ACC lesion had minimal effects on the rate of most call types during this period'. Is this period from weeks 3 to 6 relative to the proportions in week 2? I think I also don't quite understand the chord diagram. The legend says 'the numbers around each chord diagram represents relative probability value for each call type transition' so how does that relate to the proportion of these call types? It looks like there is a wider slice for cries for ACC-lesioned animals each week. I also don't see in the week 4 chord diagram, the text description of 'elevation in the rate of 'other' calls, which comprised tsik, egg, eck, chatter and seep calls. These calls were significantly elevated in animals after the ACC lesion."*

We apologize for the confusion. Fig 3D and Fig 3F are not directly related. Fig. 3D shows the different types of emitted calls. The figure shows the averaged data per group pooled from post-surgery weeks (week 3 – week 6). It represents the proportion of individual call types relative to the total number of calls during each recording period. The only major finding here was the increased rate of ‘other’ calls comprising tsik, egg, ock, chatter and seep calls. These calls were significantly elevated in animals after the ACC lesion.

While Fig. 3D represents the differences in the proportion of calls, the chord diagrams in Fig. 3F represents the probability of call-to-call transition obtained from a probability matrix. At postnatal week 6, marmosets with ACC lesions showed a higher likelihood of transitions between all call types, but less frequent transitions between social contact calls relative to sham controls. The chord diagrams visualize the weighted probabilities and directionality of these transitions between the different call types. Weighted probabilities were used to account for variations in call counts. The thickness of the arrows or links indicates the probability of a call transition, while the numbers surrounding each chord diagram represent the relative probability value for each specific transition. We have now reworded the text and clarified these details in the figure legend (pages 32-33).

(Figure 3E) How is the ratio on the y-axis calculated here?

The y-axis represents the averaged value of the ratios of the number of social contact calls relative to non-social contact calls in each recording per subject per group (i.e., x̄ (# social calls **/** # non-social calls)). This is now included in the figure legend and the axis is updated (page 32).

Also, cries could be considered a 'social contact call' since they are produced by infants to elicit responses from the parents. There is also the hypothesis in the literature that cries transition into phees.

The reviewer is correct. Cries are often considered a social contact call because they elicit parental feedback. We decided to separate cry-calls from other social contact calls for two reasons. First, in our sample, we found cry behavior to be highly variable across the animals. For example, one control infant cried incessantly whereas another control infant cried less than normal. This extreme variability in animals of the same group masked the features between animals that reliably differentiated between them. Second, cry-calls elicit feedback from parents who are normally within the vicinity of the infant whereas phee calls elicit antiphonal phee calls from any distantly located conspecific. In other words, the context in which these calls are often elicited are very different.

The use of 'syntactical' is a bit jarring to me because outside of linguistics, its use in animal communication generally refers to meaning-bearing units that can be combined into well-formed complexes such as pod-specific whale songs or predator alarm calls with concatenated syllable types in some species of monkeys. To my knowledge, individual phee syllables have not been currently shown to carry information on their own and may be better described as 'sequential' rather than 'syntactical'.

We agree. We have made this change accordingly.

(Figure 4B) How many phee calls with differing numbers of syllables are present each week? How equal is the distribution given that later analyses go up to 5 syllables?

The total number of phee calls with differing number of syllables ranged between 20-40 phees. This number varied between subjects, per week. The most common were 3- and 4-syllable phee calls which ranged from 7-15. Due to this variability, Fig. 4B presents the average syllable count. The axis is now updated.

(Figure 4C-E) How is the data combined here? Is there a 2nd syllable, the combined data from the 2nd syllable from phee calls of all lengths (1 - 5?). If so, are there differences based on how long the total sequence is?

The combined data represents the specific syllable (e.g., the 1st syllable in a 2-syllable phee, in a 3-syllable phee and in a 4-syllable phee) irrespective of the length of the sequence in a sequence. No differences were observed between 2nd syllable in a 2 syllable phee and 2nd syllable in a 3 or a 4 syllable phee. We have included this detail in the figure legend (page 33-34).

So duration is a vocal parameter that is highly dependent on physical factors such as body size and lung volume, where there differences in physical growth between the pairs of ACC-lesioned marmosets and their twins? Entropy is less closely tied to these physical factors but has previously been shown to decrease as phee calls mature, which we can also see in the negative relationship of the control animals. Do you know of experiments that show that lower entropy calls are more 'blunted'?

Thank you for raising the important issue of physical growth factors. For twin pairs, it is not uncommon for one infant to be slightly bigger, heavier or stronger than the other presumably because one gets more access to food. With increasing age, we did not observe significant changes in bodyweight between the groups. We examined grip strength in all infants as a means of assessing how well the infant was able to access food during nursing. Poor grip strength would indicate a lower propensity to ‘hang on’ to the mother for nursing which could lead to lower weight gain and reduced physical growth. We found that both grip strength and body weight increased as the infants got older and both parameters were equivalent. We have included an additional figure to show the normal increase in both weight and grip strength to the supplemental materials (Fig. S3) and have made reference to this in the text (page 8).

As for entropy, it’s impact on the emotional quality of vocalizations has not been systematically explored. Generally speaking, high entropy relates to high randomness and distortion in the signal. Accordingly, one view posits low-entropy phee calls represent mature sounding calls relative to noisy and immature high-entropy calls (e.g., Takahasi et al 2017). In the current study, the reduction in syllable entropy observed for both groups of animals with increasing age is consistent with this view. At the same time entropy can relate to vocal complexity; high entropy refers to complex and variable sound patterns whereas low entropy sounds are predictable, less diverse and simple vocal sequences (Kershenbaum, A. 2013. Entropy rate as a measure of animal vocal complexity. Bioacoustics, 23(3), 195–208). One possibility is that call maturity does not equate directly to emotional quality. In other words, a low-entropy mature call can also be lacking in emotion as observed in humans with ACC damage; these patients show mature speech, but they lack the variations in rhythms, patterns and intonation (i.e., prosody) that would normally convey emotional salience and meaning. Our observation of a reduction in phee syllable entropy in the ACC group in the context of being short and loud with reduced peak frequency is consistent with this view. Our use of the word ‘blunt’ was to convey how the calls exhibited by the ACC group were potentially lacking emotional meaning. Beyond this speculation, we are not aware of any papers that have examined the relationship between entropy and blunted calls directly. We have now included this speculation in the discussion (pages 12-13).

**Reviewer #2 (Public Review):**
The authors state that the integrity of white matter tracts at the injection site was impacted but do not show data.

We have added representative micrographs of a control and ACC-lesioned animal in a new supplementary figure which shows the neurotoxin impacted the integrity of white matter tracts local to the site of the lesion (Fig. S2).

The study only provides data up to the 6th week after birth. Given the plasticity of the cortex, it would be interesting to see if these impairments in vocal behavior persist throughout adulthood or if the lesioned marmosets will recover their social-vocal behavior compared to the control animals.

We agree. Our original intention was to examine behavior into adulthood. Unfortunately, the COVID-19 pandemic compromised the continuation of the study. We were limited by the data that we were allowed to acquire due to imposed restrictions. Some non-vocalization data collected when the animals were young adults is currently being prepared for another paper.

Even though this study focuses entirely on the development of social vocalizations, providing data about altered social non-vocal behaviors that accompany ACC lesions is missing. This data can provide further insights and generate new hypotheses about the exact role of ACC in social vocal development. For example, do these marmosets behave differently towards their conspecifics or family members and vice versa, and is this an alternate cause for the observed changes in social-vocal development?

We agree. At the time however, apparatus for assessing behavior between the infant’s family and non-family members was not available. Assessing such behaviors in the animals holding room posed some difficulty since marmosets are easily distracted by other animals as well as the presence of an experimenter, amongst other things. This is an area of investigation we are currently pursuing.

**Reviewer #3 (Public Review):**
It is striking to find that the vocal repertoire of infant marmosets was not significantly affected by ACC lesions. During development, the neural circuits are still maturing and the role of different brain regions may evolve over time. While the ACC likely contributes to vocalizations across the lifespan, its relative importance may vary depending on the developmental stage. In neonates, vocalizations may be more reflexive or driven by physiological needs. At this stage, the ACC may play a role in basic socioemotional regulation but may not be as critical for vocal production. Since the animals lived for two years, further analysis might be helpful to elucidate the precise role of ACC in the vocal behavior of marmosets.Figure 3D. According to the Introduction "...infant ACC lesions abolish the characteristic cries that infants normally issue when separated from its mother". Are the present results in marmosets showing the opposite effect? Please discuss.

To date, the work of Maclean (1985) is the only publication that describes the effect of early cingulate ablation on the spontaneous production of ‘separation calls’ largely construed as cries, coos and whimpers in response to maternal separation. All of this work was largely performed in rhesus macaques or squirrel monkeys. In addition to ablating the cingulate cortex, Maclean found that it was necessary to ablate the subcallosal (areas 25) and preseptal cingulate cortex (presumably referring to prelimbic area 32) to permanently eliminate the spontaneous production of separation cry calls. Our ablation of the ACC was more circumscribed to area 24 and is therefore consistent with MacLean’s earlier work that removal of ACC alone does not eliminate cry behavior. In adults, ACC ablation is insufficient at eliminating vocalization as well. We make reference to this on pages 13-14 of the discussion.

Figure 3E and Discussion. Phees are mature contact calls and cries immature contact calls (Zhang et al, 2019, Nat Commun). Therefore, I would rather say that the proportion of immature (cries) contact calls increases vs the mature (phee, trill, twitters) contact calls in the ACC group. Cries are also "isolated-induced contact calls" to attract the attention of the caregivers.

The reviewer is correct in that cries are directed towards caregivers but in our sample, cry behavior was highly variable between the infants. Consequently, in Fig. 3E social contact calls include phee, twitter and trill calls but does not include cries which were separated (see also response to reviewer #1). Many of the calls made during babbling were immature in their spectral pattern (compare phee calls between Fig. 3A and 3B). Cries typically transitioned into phees, twitters or trills before they fully matured. Fig 3E shows that the controls made more isolation-induced social contact calls at postnatal week 6 which were presumably maturing at this time point. Thus, if anything, there was an increase in the proportion of mature contact calls vs immature contact calls with increasing age.

Figure 4D. Animal location and head direction within the recording incubator can have significant effects on the perceived amplitude of a call. Were these factors taken into account?

The reviewer makes an excellent observation. Unfortunately, we did not account for location and head direction because the infants were quite mobile in the incubator. The directional microphone was hidden from view because the infants were distracted by it, and positioned ~12 cm from the marmoset, and placed in the exact same location for every recording. In addition, calls with phantom frequencies were eliminated during visual inspection of spectrograms. Beyond these details, location and head direction were not taken into account.

Figure 4E. When a phee call has a higher amplitude, as is the case for the ACC group (Figure 4D), the energy of the signal will be concentrated more strongly at the phee call frequency ~8KHz. This concentration of the energy reduces the variability in the frequency distribution, leading to lower entropy. The interpretation of the results should be reconsidered. A faint call (control group) can exhibit more variability in the frequency content since the energy is distributed across a wider range of frequencies contributing to higher entropy. It can still be "fixed, regular, and stereotyped" if the behavior is consistent or predictable with little variation. Also, to define ACC calls as "monotonic" I would rather search for the lack of frequency modulation, amplitude variation, or narrower bandwidth.

We very much appreciate this explanation. We were able to identify the maximum frequency that closely matched pitch of a sound for each syllable in a multisyllabic phee. New Fig. 4E shows that the peak frequency for each phee syllable was lower in the ACC-lesioned monkeys which may directly translate to the low entropy observed in this group. The term “monotonic” was used to relate our data to the classical and long-standing evidence of human ACC lesions causing monotonous intonation of speech. When all factors are taken into account, it is evident that the vocal phee signature of the ACC-lesioned animal was structurally different to the controls implicating a less complex and stereotyped ACC signal. Further studies are needed to systematically explore the relationship between entropy and emotional quality of vocalizations

Apart from the changes in the vocal behavior, did the AAC lesions manifest in any other observable cognitive, emotional, or social behavior? ACC plays a role in processing pain and modulating pain perception. Could that be the reason for the observed increase in the proportion of cries in the ACC group and the increase in the phee call amplitude? Did the cries in the ACC group also display a higher amplitude than the cries in the control group?

It was our intention to acquire as much data as possible from these infants as they matured from a cognitive, social and emotional perspective. Unfortunately, our study was hampered by variety of reasons including the COVID-19 pandemic which imposed major restrictions on our ability to continue with the experiment in a time sensitive manner. In addition, the development and construction of the custom apparatus to measure these behaviors was stalled during this period further preventing us from collecting behavioral data at regular time intervals. As for the cry behavior, the number of cries, in the ACC group were very low especially at postnatal week 5 and 6. Consequently, there were very few data points to work with.

Discussion. Louder calls have the potential to travel longer distances compared to fainter calls, possess higher energy levels, and can propagate through the environment more effectively. If the ACC group produced louder phee syllables, how could be the message conveyed over long distances "deficient, limited, and/or indiscriminate"?

Thanks for raising this interesting concept. Not all calls emitted by the animals were loud. We specifically examined the long-distance phee call in this regard. The phee syllables emitted by the ACC group were high amplitude with low frequencies, short duration and low entropy. Taking these factors into account, it is conceivable that the phee calls produced by the ACC group could not effectively convey their message over long distances despite their propagation through the environment. We have made reference to this in the discussion where we focus is specifically on the phee calls only (pages 12).

Abstract: Do marmosets have syntax? Consider replacing "syntactical" with a more appropriate term (maybe "syntax-like").

Thanks for this suggestion. We have replaced the term syntactical with ‘sequential’ as per the recommendation of reviewer #1.

Introduction: "...cries that infants normally issue when separated from its mother". Please replace "its" with "their".

This has been corrected.

Results: Is the reference to Fig 1B related to the text?

We have included and referred to Fig. 1B in the text (results and methods) to show other researchers how they can use this technique as a reliable and safe means of monitoring tidal volume under anesthesia in small infant marmoset without intubation.

I understand that both "spectrograph" and "spectrogram" are used to analyze the frequency content of a signal. Nevertheless, "spectrogram" refers to the visual representation of the frequency content of a signal over time, and this term is commonly used in audio signal processing and specifically in the vocal communication field. I would recommend replacing "spectrograph" with "spectrogram".

Thanks for this suggestion. We have corrected this throughout the manuscript.

(Concerning the previous comment in the public review). Cries are uttered to attract the attention of the caregivers. The increase in the proportion of cries in the ACC group does not match the sentence: "...these infants appeared to make little effort in using vocalizations to solicit social contact when socially isolated".

We apologize for the confusion. It is not the case that the ACC animals make more cries. Cry calls were highly variable amongst the animals. Consequently, although Fig 3D gives the impression that the proportion of cries in higher in ACC animals they did not differ significantly from the controls. Due to their high variability, cries were removed in the measurement of social contact. Accordingly, Fig. 3E does not include cry behavior; it shows that the ACC animals engage less in social contact calls.

Related to Figure 3. What is the difference between "egg" and "eck" calls? Do you mean "ock"?

We apologize. This was a typo. It should be ock calls.

Figure 4B. Is the sample size five animals per group and per week? Overlapping data points seem to be placed next to each other. Why in some groups (e.g. ACC 6 weeks) less than five dots are visible?

The sample size differed per week because of the lack of recording during the COVID restrictions. In Fig 4b, we have now separated the overlapping dots. We have also added the sample size of the groups in the figure legends.

Would the authors expect to see stronger differences between the lesioned and the control groups when comparing a later developmental stage? The animals were euthanized at the age of

These speculation is certainly feasible and yes, we were hoping to establish this level of detail with testing at later developmental stages. This is an aspect of development we are currently pursuing.

Could these experiments be conducted?

I’m afraid these animals are longer available, but we are currently conducting experiments in other animals with early life neurochemical manipulations who show behavioral changes into early adulthood.

ACC lesion: It is reported that the lesions extended past 24b into motor area 6M. Did the animal display any motor control disability?

Surprisingly, despite the lesion encroaching into 6M, these animals showed no observable motor impairment. We assessed the animals grip strength and body weight and discovered normal strength and growth in weight in both controls and the lesioned group. We have added this data as supplemental information (Fig. S3).